# Targeted Next-Generation Sequencing Analysis Predicts the Recurrence in Resected Lung Adenocarcinoma Harboring *EGFR* Mutations

**DOI:** 10.3390/cancers13143632

**Published:** 2021-07-20

**Authors:** In Ae Kim, Jae Young Hur, Hee Joung Kim, Song Am Lee, Jae Joon Hwang, Wan Seop Kim, Kye Young Lee

**Affiliations:** 1Precision Medicine Lung Cancer Center, Konkuk University Medical Center, Seoul 05030, Korea; 20180618@kuh.ac.kr (I.A.K.); 20160475@kuh.ac.kr (J.Y.H.); hjkim@kuh.ac.kr (H.J.K.); wskim@kuh.ac.kr (W.S.K.); 2Department of Pathology, Konkuk University School of Medicine, Seoul 05030, Korea; 3Department of Pulmonary Medicine, Konkuk University School of Medicine, Seoul 05030, Korea; 4Department of Thoracic Surgery, Konkuk University School of Medicine, Seoul 05030, Korea; azzy@kuh.ac.kr (S.A.L.); hjj@kuh.ac.kr (J.J.H.)

**Keywords:** targeted next-generation sequencing, resected *EGFR* mutated-lung adenocarcinoma, *CTNNB1*, post-operative recurrence, recurrence-free survival

## Abstract

**Simple Summary:**

Despite complete resection and adjuvant chemotherapy, the recurrence rate of early *EGFR*-mutated lung adenocarcinoma remains high. We hypothesized that patients with recurrence-related genetic alterations would have poor prognosis and analyzed the genetic profiles of 131 patients using next-generation sequencing (NGS) with a 207 cancer-related gene panel. As a result, we revealed several negative prognostic factors for recurrence, such as a large number of concomitant mutations and the existence of specific mutation subtypes. Targeted NGS analysis provides information on the prognosis of patients with resected EGFR-mutation lung adenocarcinoma and helps to identify patients with high relapse risks who require intensive chemotherapy or adjuvant *EGFR*-TKIs treatments.

**Abstract:**

Targeted NGS, widely applied to identify driver oncogenes in advanced lung adenocarcinoma, may also be applied to resected early stage cancers. We investigated resected *EGFR*-mutated lung adenocarcinoma mutation profiles to evaluate prognostic impacts. Tissues from 131 patients who had complete resection of stage I–IIIA *EGFR*-mutated lung adenocarcinoma were analyzed by targeted NGS for 207 cancer-related genes. Recurrence free survival (RFS) was estimated according to genetic alterations using the Kaplan–Meier method and Cox proportional regression analysis. The relapse rate was 25.2% (33/131). Five-year RFS of stages IA, IB, II, and IIIA were 82%, 75%, 35%, and 0%, respectively (*p* < 0.001). RFS decreased with the number of co-mutations (*p* = 0.025). Among co-mutations, the *CTNNB1* mutation was associated with short RFS in a multivariate analysis (hazard ratio: 5.4, 95% confidence interval: 2.1–14.4; *p* = 0.001). *TP53* mutations were associated with short RFS in stage IB–IIIA (*p* = 0.01). RFS was shorter with *EGFR* exon 19 deletion (19-del) than with mutation 21-L858R in stage IB–IIIA tumors (*p* = 0.008). Among 19-del subtypes, pL747_P753delinS (6/56, 8.9%) had shorter RFS than pE746_A750del (39/56, 69.6%), the most frequent subtype (*p* = 0.004).

## 1. Introduction

Complete surgical resection is the standard treatment for early stage lung cancer. However, the rate of recurrence is high and patients with recurrence eventually die of disease progression. Although epidermal growth factor receptor (EGFR) mutations are a good prognostic factor in resected lung cancer [1,2,3], many patients with early resected *EGFR*-mutated lung cancer experience recurrence [4]. The recurrence rate of patients with stage IB tumors is 20%, which is less than those of patients with stage II (40–60%) and IIIA (70–100%) tumors [5]. However, tumor-stage distribution analysis shows that stage I tumors with low recurrence rates are much more prevalent than stage II and IIIA tumors, thus the number of patients with recurrence in stage I is not negligible [6]. Therefore, it is an important unmet need to identify patients at high risk for recurrence in resected early stage *EGFR*-mutated lung adenocarcinoma.

EGFR-tyrosine-kinase inhibitors (TKIs) have higher efficacy and lower toxicity in advanced settings. EGFR-TKIs have provided longer progression-free survival than adjuvant chemotherapy [7] but have not shown better overall survival than chemotherapy in the adjuvant setting [8,9]. Recently, adjuvant treatment of resected stage IB–IIIA *EGFR*-mutated non-small cell lung cancer (NSCLC) with the 3rd generation EGFR-TKI osimertinib in the ADAURA clinical trial showed significantly better outcomes than that of the placebo [10]. However, considering the long treatment period (three years) and high cost, it is questionable whether all stage IB NSCLC should be treated with EGFR-TKISs, despite the low recurrence rate [9,11]. Therefore, appropriate patient selection is important when considering adjuvant TKI therapy.

Targeted next-generation sequencing (NGS) is widely applied in personalized therapy for patients with NSCLC in order to identify driver mutations. This allows the generation of comprehensive genetic profiles in one assay, identifying multiple oncogenic alterations missed in previous conventional assays. Recent NGS studies have shown that *EGFR* mutations and co-mutations appear to influence the prognosis of advanced *EGFR*-mutated lung cancers [12,13,14]; however, there have been only a few targeted NGS analysis studies regarding the prognosis of resected early stage *EGFR*-mutated lung cancer that focused on recurrence [15,16,17].

In the current study, we analyzed the genetic profiles of 131 patients with stage I–IIIA *EGFR*-mutated lung adenocarcinoma tumors using targeted NGS. We evaluated the risk of recurrence based on co-occurring actionable mutations, the number of co-mutations, *EGFR* mutation types, and *EGFR* exon 19 deletion (19-del) subtypes in resected *EGFR*-mutated lung adenocarcinomas. We believe our results will help in identifying patients at high risk of recurrence who may benefit from adjuvant EGFR-TKIs therapy.

## 2. Materials and Methods

### 2.1. The Patients and Specimen Collection

We enrolled 131 patients histologically confirmed with stage I to stage IIIA *EGFR*-mutated lung adenocarcinoma who underwent surgical resection at Konkuk University Medical Center from September 2005 to May 2017. All specimens underwent *EGFR* genetic testing and were analyzed by targeted NGS using a panel that included 207 cancer-related genes (Appendix A). Cancer stage was classified according to 8th American Joint Committee on Cancer (AJCC) criteria [18]. The medical records of all enrolled patients were retrospectively reviewed. In addition to pathologic data, age, sex, smoking history, stage, surgical data (surgery date, methods, and extent of resection), and dates of recurrence and death were collected. Patients who had another cancer and received any neo-adjuvant treatment were excluded. The study protocol was approved by the Konkuk University Medical Center Institutional Review Board (approval number: KUH 1210049) and the need for written informed consent from the participants was waived due to the retrospective nature of this study and the fact that the study posed no potential harm to the patients.

### 2.2. NGS Processing

DNA was extracted from formalin-fixed, paraffin-embedded tissues of the 131 patients with pulmonary adenocarcinoma using the QIAamp DNA kit (Qiagen, Hilden, Germany), according to the manufacture’s protocol. After extraction, the DNA was evaluated by targeted NGS for 207 cancer-related genes (KF1 Panel, Appendix A) using a Custom Cancer Panel v2.1 (Agilent Technologies, Inc., Santa Clara, CA, USA). Briefly, 200 ng of genomic DNA was fragmented with a Covaris E220 Focused-ultrasonicator (Covaris, Woburn, MA, USA) and subsequently subjected to end repair, tailing, and adapter ligation. Un-ligated adaptors were removed using Agencourt AMPure XP beads (Beckman Coulter, Beverly, MA, USA). The resulting libraries were polymerase chain reaction (PCR)-amplified, purified using Agencourt AMPure XP beads, and sequenced on an Illumina HiSgeq2500 platform using an average sequencing depth of 1000×. Matched germline DNA of the patients, which is a normal control for mutation analysis, was unavailable for this retrospective study.

### 2.3. NGS Data Analysis

Raw sequencing data were processed and variants were called using the Macrogen Inc. bioinformatics pipeline. The detailed NGS processing and NGS pipeline flow chart are described in the Appendix A. Somatic mutations were identified, including single nucleotide variants (SNVs), small insertions and deletions (Indels), gene rearrangements, and copy number variations (CNV). Pathogenic somatic mutations with variant allele frequencies (VAF) over 2% were regarded as significant actionable mutations and used for analyses. Macrogen Inc. guarantees the limit of detection to 1% VAF from the quantitative multiplex reference standard test. We set the cut-off value at 2% to avoid false positive detection of the mutations and intrinsic errors of NGS. The clinically significant mutations were not changed whether the value was set to either 1% or 2%.

We reduced false positive mutations by collecting tissue samples with tumor contents more than 20%, because sequence artifacts are reduced with the number of input DNA templates. To prevent artificial cytosine deamination by formalin, we fixated tissues within 24 h after we had obtained them. We used the bioinformatics program, Mutect2 to filter the oxoG artifact (G:C > T:A) form sequencing data sets.

### 2.4. Patient Follow-Up

The patients were examined at two-month intervals on an outpatient basis. The follow-up evaluation included physical examination, chest radiography, blood analysis, and computed tomography (CT) scans of the chest. Further evaluations were performed to detect additional symptoms or signs of tumor recurrence, including positron emission tomography-computed tomography (PET-CT), brain magnetic resonance imaging (MRI), and CT scans of the chest and abdomen. We diagnosed tumor recurrence based on histological confirmation of findings from physical examinations and diagnostic imaging. Second primary lung cancer was differentiated from that of recurrent NSCLC according to the criteria proposed by Martini and Melamed [19]. The date of recurrence was defined as the date of histologic proof. For patients whose diagnoses were based on clinico-radiologic findings, date of recurrence was defined as the date of identification by the physician.

### 2.5. Statistical Analysis

Clinical and pathological parameters of the patients were evaluated using χ^2^ analysis (sample size > 5) or Fisher exact test (sample size ≤ 5) for categorical variables because Fisher’s exact test is suitable when the expected values in any of the cells of a contingency table are below 5. The *t*-test and Wilcoxon rank test were used to analyze continuous variables. Recurrence-free survival (RFS) was defined in surviving patients as the time from surgery to recurrence or to last follow-up. RFS percentages were calculated using the Kaplan–Meier method and differences in RFS according to clinicopathologic factors or molecular factors were tested using the log-rank test in the univariate analysis. To identify the prognostic factor for recurrence in multivariate analysis, the Cox proportional hazards model was used to test the effect of a mutation subtype adjusted for multiple clinical/pathological factors (sex, age, smoking status, stage, surgical method, and adjuvant chemotherapy). The Kaplan–Meier method is the most common statistical method to assess the risk factor of disease recurrence and it has been widely used in previous similar studies [20,21,22]. It is especially useful when the data include cases with observation time shorter than the duration of the study [23], thus, it is the appropriate method for our study, which has low frequency events such as recurrence and requires a long observation time. For all calculations, the tests were two-sided, and significance was set at 5%. Statistical analyses were performed using Statistical Package for the Social Sciences (SPSS) for Windows version 25 software (SPSS, Inc., Chicago, IL, USA).

## 3. Results

### 3.1. Patient Characteristics

This study included 131 patients with stage I to stage IIIA EGFR-mutated lung adenocarcinoma whose tumors were resected from 2005 to 2017. Recurrence was observed in 33 of the 131 patients (25.2%). The medial follow-up period was 50.1 months. Overall, 45.8% of the patients were older than 65 years, 43.8% were male, and 37.5% were smokers. Most patients (84.6%) underwent standard lobectomy and mediastinal lymph node dissection, 20% had visceral-pleural invasion (VPI) tumors, and 5.3% had lympho-vascular invasion tumors. Relapse was not associated with sex, smoking history, or surgical procedure. However, higher stage (*p* = 0.001) and pathologic invasions, including VPI (*p* = 0.001) and lympho-vascular invasions (*p* = 0.002), were related with recurrence (Table 1). Patient treated with adjuvant chemotherapy experienced recurrence at a greater frequency (47.2% vs. 14.9%; *p* = 0.001). Of the 20 patients with stage II–IIIA disease, 16 had received adjuvant chemotherapy and 15 experienced recurrences. The adjuvant cytotoxic chemotherapy being provided to patients more susceptible to relapse was not effective in preventing recurrence.

### 3.2. Genetic Landscape of Resected EGFR-Mutated Adenocarcinoma According to Recurrence Status

Among the co-occurring mutations in *EGFR*-mutated adenocarcinoma, *TP53* mutations were the most frequently observed (16.2%), followed by *CTNNB1* mutations (4.6%) and *PIK3CA* mutations (4.6%). *RET, APC, ATM*, and *PTCH1* mutations were also found at a frequency of approximately 2% each (Appendix A). The 16.2% frequency of *TP53* mutations was lower than that of previous reports [24,25]. Previous studies have reported that higher stage tumors have more *TP53* mutations [26]. In our cohort, most tumors were in stage I (80%), thus the frequency of *TP53* mutations was lower than those in other reports. We compared the frequency of all genetic alterations to recurrence status. Patients with recurrence had *CTNNB1* mutations at a significantly greater frequency than patients without recurrence (60.9% vs. 34.7%; *p* = 0.001; Figure 1).

### 3.3. Cancer Stage Distribution of Resected EGFR-Mutated Lung Adenocarcinoma and Recurrence

The proportion of patients in our cohort with stage IA and stage IB tumors were 62.6% and 22.1%, respectively, while those with stage II and stage IIIA were only 6.9% and 8.4%, respectively (Table 1). In other words, the majority of resected lung adenocarcinomas are at stage IA of the disease. The recurrence rate of 75% (15/20) for stage II–IIIA was higher than 15.3% (17/111) for stage I. However, the number of patients with stage I disease that experienced recurrence was notable, even though the recurrence rate of stage I disease is lower than those of stages II–IIIA disease.

Patients with *EGFR*-mutated lung adenocarcinoma of more advanced stage in our cohort had shorter RFS than that previously reported (*p* < 0.001; Figure 2a). Notably, RFS was similar in stage II and stage IIIA adenocarcinoma.

### 3.4. Association between the Number of Genetic Alterations and Recurrence

We investigated the distribution of the number of genetic alterations in resected *EGFR*-mutated adenocarcinoma. Of the patients with *EGFR*-mutated lung adenocarcinoma, 52.3% had no mutations in addition to the *EGFR* mutation, 33.9% had one concomitant mutation, and 10% had more than two concomitant mutations. Detailed information regarding the concomitant mutations is summarized in Appendix A. We further analyzed RFS based on the number of concurrent mutations. RFS times became short as the number of concurrent mutations increased. Tumors with fewer concurrent mutations were associated with longer RFS (*p* = 0.015; Figure 2b). Furthermore, EGFR-mutated lung adenocarcinoma with concurrent mutations was more likely to recur.

### 3.5. Recurrence Rate According to EGFR Mutation Subtypes and Co-Mutations

We investigated recurrence rates according to *EGFR* mutation subtypes. Relapse rates decreased 33.3% (5/15) for other *EGFR* mutations, 30.3% (17/56) for 19 -del and 18.6% (11/60) for 21L858R (Figure 3a,b), but there was no statistical difference among the patients as a whole (*p* = 0.22). Though the *p* value does not indicate statistical difference, the graph in Figure 3b indicates there exists a difference of recurrence rates depending on *EGFR* types. Thus, we analyzed the recurrence rates excluding stage IA because stage IA occupy a large proportion of the patients as a whole and their recurrence rates were low. In this analysis, shown in Figure 3c, the RFS difference between 19-del and 21-L858R was obvious in the subgroup analysis of the patients with stage IB–IIIA cancer. If we excluded all stage IA and IB patients from the analysis, only 20 patients would be left, which makes any analysis difficult. If the total population of the analysis becomes larger by accumulating more data, the tendency will become clearer.

These patients with 19-del had lower RFS than those with the exon 21-L858R mutation (Figure 3c; *p* = 0.008). The 19-del mutation was a more potent oncogenic variant than that of the exon 21-L858R point mutation. To confirm the tendency, we analyzed the RFS of the patients with 3 frequent *EGFR* mutations (E746_A750del, L747_P753delinS of 19-Del mutation and 21L858R) in whole patients in Figure 3d.

A total of 56 patients had a 19-del mutation (56 of 131, 42.7%). Of the patients with 19-del, 69.6% (39 of 56) had the pE746_A750del mutation while 10.7% (6 of 56) had the pL747_P753delinS mutation. While the most common type of 19-del was pE746_A750del, the recurrence rate was higher for pL747_P753delinS than that of pE746_A750del. (*p* = 0.001; Table 2, Figure 3d) [24,25]. The detailed frequency of *EGFR* mutation and recurrence rate are in Appendix A.

We also investigated recurrence related factors in terms of co-mutations. Consistent with previous reports [26,27], we observed the *TP53* mutations were poor RFS-related factors for recurrence in stage IB–IIIA tumors with *EGFR*-mutated lung adenocarcinoma (Figure 3e; *p* = 0.01). Notably, the *CTNNB1* mutation was associated with shorter RFS (Figure 3f; *p* < 0.001). *TP53* mutations have seldom been shown to be an indicator of prognosis in previous research, but our current results clearly indicate *CTNNB1* may be a recurrence-related gene that we should be cognizant of.

### 3.6. Prognostic Factors for Recurrence Based on Multivariate Analysis of Resected EGFR-Mutated Lung Adenocarcinomas

Univariate analysis of RFS revealed the number of concomitant mutations, CTNNB1 co-mutation, and 19-del subtype, especially, pL747_P753delinS, were negative prognostic factors for recurrence. We also evaluated recurrence-related factors using multivariate analysis adjusted by age, sex, stage, smoking history, operational method, and adjuvant chemotherapy. We found that sex, smoking history, and surgery method were not risk factors, contrary to our expectations. However, higher stage, the presence of VPI, and higher numbers of co-mutations were related with recurrence according to the multivariate analysis. In terms of EGFR mutation subtypes, 19-del (especially pL747_P753delinS) was related to recurrence in the multivariate analysis. Among other co-mutations, the CTNNB1 mutation was an independent risk factor (hazard ratio 8.65; 95% confidence interval 3.0–24.9; *p* < 0.001) and TP53 (hazard ratio 3.48; *p* = 0.02) was a risk factor for recurrence in patients with stage IB–IIIA cancer (Table 2).

## 4. Discussion

Despite complete resection and adjuvant chemotherapy, the recurrence rate of *EGFR*-mutated lung adenocarcinoma exceeds expectations, highlighting the need for improved treatment strategies. We hypothesized that patients with larger numbers of recurrence-related genetic alterations would have poor prognosis and need more intensive chemotherapy and surveillance. Targeted NGS is able to provide information on concurrent mutations, as well as on targeted driver mutations. Therefore, we analyzed the genetic profiles of 131 patients with stage I to stage IIIA *EGFR*-mutated lung adenocarcinoma using NGS and compared the risk of recurrence based on co-occurring actionable mutations, the number of co-mutations, *EGFR* mutation types, and 19-del subtypes. We believe this work is the first NGS study focusing on resected *EGFR*-mutated adenocarcinoma that compared recurrence rates.

In the analysis of clinical-pathologic characteristics of our cohort, the ratios of male/female, young/old, and smoker/non-smoker were almost the same. However, there was an imbalance in stage distribution with the proportion of stage I lung adenocarcinoma (85%) being much greater than that of stage II–IIIA (15%). The patients in our study who were treated by adjuvant chemotherapy experienced a greater rate of recurrence. We believe this was due to the adjuvant cytotoxic chemotherapy being more often provided to patients with high risk factors for relapse. Regardless of the reason, these results suggest that adjuvant cytotoxic chemotherapy does not effectively prevent recurrence. Thus, other treatment strategies are needed to prevent relapse.

We classified the patients in our cohort into four groups based on cancer stage, IA, IB, II, and IIIA and then compared RFS of the groups. Tumors of more advanced stage were associated with shorter RFS. The prognosis of stage II was poor and the RFS for this group was similar to that of the stage IIIA group. Based on these results, EGFR-mutated stage II lung adenocarcinoma should not be treated with conventional surgery plus adjuvant chemotherapy, but the adjuvant or neoadjuvant treatment should include TKIs. 

We also explored the molecular profiles associated with recurrence and found the number of co-occurring mutations was related with recurrence. The greater the number of co-occurring mutations, the shorter the RFS. The accompanying concurrent mutations clearly provide an impact on recurrence in resected *EGFR*-mutated lung adenocarcinoma as the EGFR- mutated lung adenocarcinoma with more concurrent mutations had poorer response rate to EGFR-TKIs in advanced stages [12,13].

We also found in our multivariate analysis that *CTNNB1* mutations co-occurring with *EGFR* mutations were independent negative predictive factors for recurrence. We examined the literature explaining the reason why most patients with *CTNNB1* mutations encoding β-catenin, had recurrence after the complete tumor resection. It is suggested by accumulating data that the Wnt/β-catenin pathway is related to tumorigenesis and metastasis of lung cancer. In the normal state, β-catenin is degraded by a complex consisting of adenomatous polyposis coli, axin, and glycogen synthase kinase 3b. If β-catenin is not degraded by the aberration of Wnt/β-catenin signaling, it remains in the cytoplasm. The increased β-catenin in the cytoplasm moves to the nucleus, acts as a transcription factor to activate cell cycles continuously, and it induces tumor formation [28,29]. In our study, relapses of tumors with *CTNNB1* mutations usually occurred in distant organs (two cases brain, one case adrenal gland, one case multiple contralateral lung nodules, and one case pleural effusion). Considering that *CTNNB1* mutations were related to metastasis, they were thought to be related to micrometastasis undetected on radiologic or pathologic examination at the time of surgery.

We tried to validate the high recurrence rate of patients with *CTNNB1* in TCGA data. However, it provides the survival data only, not including recurrence related data. The frequency of *CTNNB1* was rare in public genomic data of Caucasian lung cancer cohorts. Therefore, we cannot obtain enough data to conduct an analysis from public databases. However, poor prognosis of the *CTNNB1* mutation in lung cancer has been proven in other studies. Sapiro et al. mentioned that Wnt/*CTNNB1* pathway activation increased the risk of tumor recurrence in patients with stag I NSCLC using immunoblot assay [30]. Chun et al. mentioned that the somatic mutations of *APC, CTNNB1*, and *AMER1* in the *WNT* signaling pathway were highly associated with shortened disease-free survival in 201 lung adenocarcinoma patients using NGS analysis [31]. *CTNNB1* mutation was more frequent in endometrial cancer (16%), HCC (12%), or colon cancer (6%) than NSCLC (3%) [32]. A similar result was also provided in endometrial cancer in an NGS study in which 53 of 245 patients had the *CTNNB1* mutation. In this study, the *CTNNB1* mutation was the significant biomarker of recurrence in low grade, early stage endometrial cancer (HR 4.69) [33]. Therefore, *CTNNB1* is a biomarker of recurrence undoubtedly, though the number of cases with *CTNNB1* in our study is limited due to the rarity of that mutation in lung cancer. Clinicians and investigators should pay attention to the *CTNNB1* mutation in the future.

*TP53* mutations at 16% were the most prevalent mutations in our study; however, the prevalence was low compared with that of other studies [13,34]. This may be explained by the fact that most of the patients in our study (84.7%) had stage I lung adenocarcinoma and early stage tumors are reported to have fewer *TP53* mutations than advanced-stage tumors [35]. *TP53* is a tumor suppressor gene and stage IB–IIIA tumors in our study with TP53 mutations were associated with short RFS.

In subgroup analysis of stage IB–IIIA tumors with high recurrence rates, 19-del subtype mutations were associated with shorter RFS than that of the exon 21-L858R mutation (*p* = 0.008), while 19-del in advanced-stage lung cancer has been shown to be a good predictive factor for the use of EGFR-TKIs [36,37]. 19-del in early stage resected lung cancer may be a poorer prognosis factor than that of the exon 21-L858R mutation. We also explored the impact of 19-del subtypes on recurrence and found the 19pE746_A750deletion mutation was more frequently detected than that of the pL747_P753delinS mutation. However, the pL747_P753delinS mutant was associated with a higher rate of recurrence than that of the pE746_A750del mutant. The pE746_A750del mutation has a 5 amino acid deletion (amino acids 746 to 750) while pL747_P753delinS has a 6 amino acid net change with 7 amino acids being deleted (amino acids 747 to 753) and a serine being inserted. We hypothesize the recurrence rate may be associated with the extent of amino acid changes. Multiple amino-acid changes, such as that of 19-del, would be more likely to cause recurrence than that of a single amino-acid change, such as that of the exon 21-L858R point mutant. In the 19-del subtypes, the 6 amino acid change was associated with more recurrence than that of the 5 amino acid change. We believe the greater number of amino acid changes resulted in greater structural change of the EGFR protein and thereby increased its oncogenic activity [24,25,38].

Additionally, we compared the RFS between *EGFR* mutated cases and wild type cases (*EGFR*^wt^). There is a tendency that patients with *EGFR* mutations have better prognosis than those with *EGFR* wild type (*p* = 0.05) (Appendix A). Furthermore, we analyzed the four sub-groups, *EGFR*^wt^, *EGFR*^L858R^, *EGFR*^E746_A750del^, and *EGFR*^L747_P753delinS^ and the results are shown in Appendix A. *EGFR*^L858R^ showed better RFS than that of *EGFR*^wt^. The prognosis of *EGFR*^wt^ was similar to *EGFR*^E746_A750del^. However, there was no difference in the number of co-mutations among subgroups.

One of the strong points in our current study was that it did not include physician selection bias as we included specimens from most of the EGFR-mutated lung adenocarcinomas resected at our institution over nearly a 12-year period. Furthermore, NGS sequencing was performed at an outside company, Macrogen Inc., where the NGS pipeline is well established. Another strong point of our study was that it included a homogenous cohort of patients with EGFR-mutated lung adenocarcinoma. This is in contrast to previous studies that included a mix of patients with NSCLC, squamous cancer, and wild-type adenocarcinoma. However, a weakness in our study is the fact that it was a retrospective study of from a single treatment center. As a result, we did not obtain the blood or normal tissue to exclude germline. Our findings should be verified in a multicenter study involving a larger study cohort. Paired tumor-normal tissue NGS testing is the best strategy to exclude germline mutation, but double cost limits its utility only in investigation. Thus, tumor-only mutation detection is performed in real clinical setting and it results in false positive detection. To identify the germline mutations without blood samples, an ordinal filtering approach should be performed with information from variant population databases. It is necessary to develop a large public tumor bank for public databases and an accurate and validated germline mutation filtration tool.

In our study, the observation period for postoperative recurrence of EGFR mutation-positive lung cancer is not consistent. Thus, we used the Kaplan–Meier curve, taking into account censored data that occur when a patient is lost to follow-up or alive without recurrence at the last follow-up. The Kaplan–Meier curve shows the sum (proportion) of the alive patients at each time point excluding events such as death or recurrence. We analyzed recurrence-free survival time depending on EGFR mutation, and the difference of RFS in each group was analyzed by the log-rank test. In addition, the long observation period is important to reduce the possibility of later recurrence. Thus, the median follow up time of more than four years makes our data reliable because most recurrence occurs within two or three years after surgery [39].

The United States Food and Drug Association recently approved the 3rd generation EGFR-TKI osimertinib as an adjuvant treatment for use following the resection of *EGFR*-mutated lung cancer. The 2-year DFS rate was 89% for the osimertinib-treated group vs. 53% for the placebo group (hazard ratio: 0.21; *p* < 0.0001). However, considering the high cost and adverse effect of TKIs, the selection of patient who have a high risk of relapse is important. Targeted NGS analysis can provide comprehensive molecular information that can be used to distinguish patients with a high genetic risk of relapse. Therefore, the identification of EGFR mutation subtypes and other concurrent mutations may lead to improved patient survival by optimizing treatment regimens. In addition, considering the poor clinical outcomes of the patients with *CTNNB1* mutation, targeted therapy or chemotherapy should be provided. Additionally, patients with *CTNNB1* mutation need intensive surveillance for the early detection of recurrence due to high relapse risk. Targeting β-catenin pathway therapy in clinical trials should be done in the future, as well [40]. Further prospective clinical studies need to discuss what treatment is beneficial to patients with the *CTNNB1* mutation.

## 5. Conclusions

Targeted NGS analysis revealed negative prognostic factors for recurrence. The factors included the number of concomitant mutations, the existence of *CTNNB1* co-mutations, and the existence of 19-del subtypes, especially pL747_P753delinS. Targeted NGS analysis helped predict the prognosis and recurrence of patients with resected *EGFR*-mutation lung adenocarcinoma and benefitted in the identification of patients with a high risk of recurrence for selection for adjuvant EGFR-TKIs treatment.

## Figures and Tables

**Figure 1 cancers-13-03632-f001:**
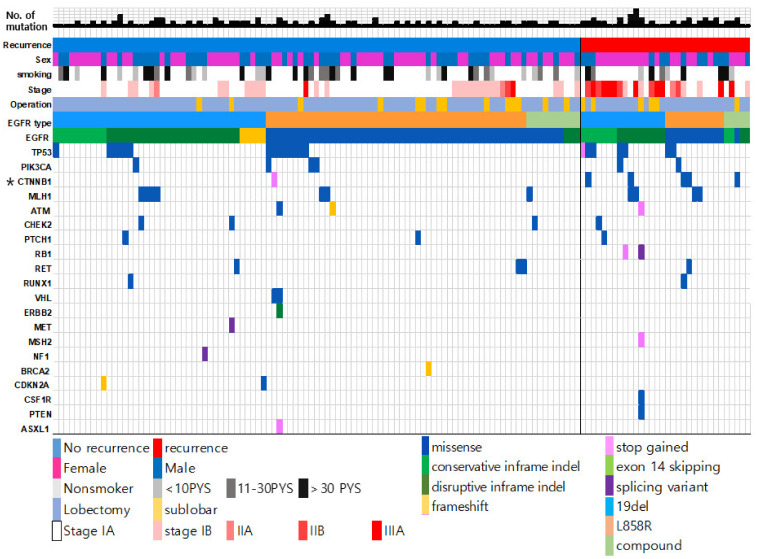
Genetic landscape of resected EGFR–mutated lung adenocarcinoma. The mutation data from all the patients in this study are included. The middle vertical line divides the patients into two groups: recurrence case (**right**) and non-recurrence case (**left**).

**Figure 2 cancers-13-03632-f002:**
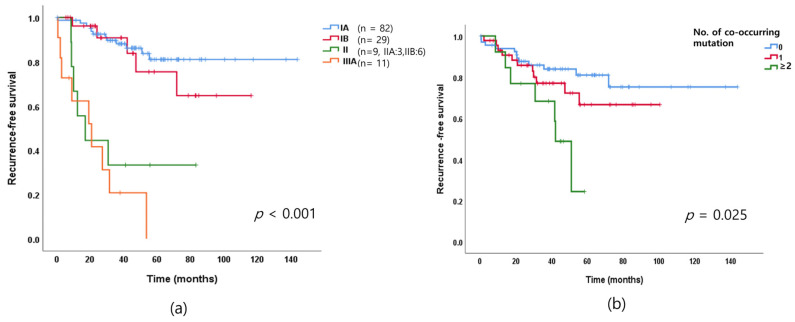
(**a**) Recurrence-free survival (RFS) in *EGFR*-mutated lung cancer according to (**a**) cancer stage and (**b**) number of co-occurring mutations. The RFS was significantly shorter according to the number of mutations co-occurring with the *EGFR* mutation.

**Figure 3 cancers-13-03632-f003:**
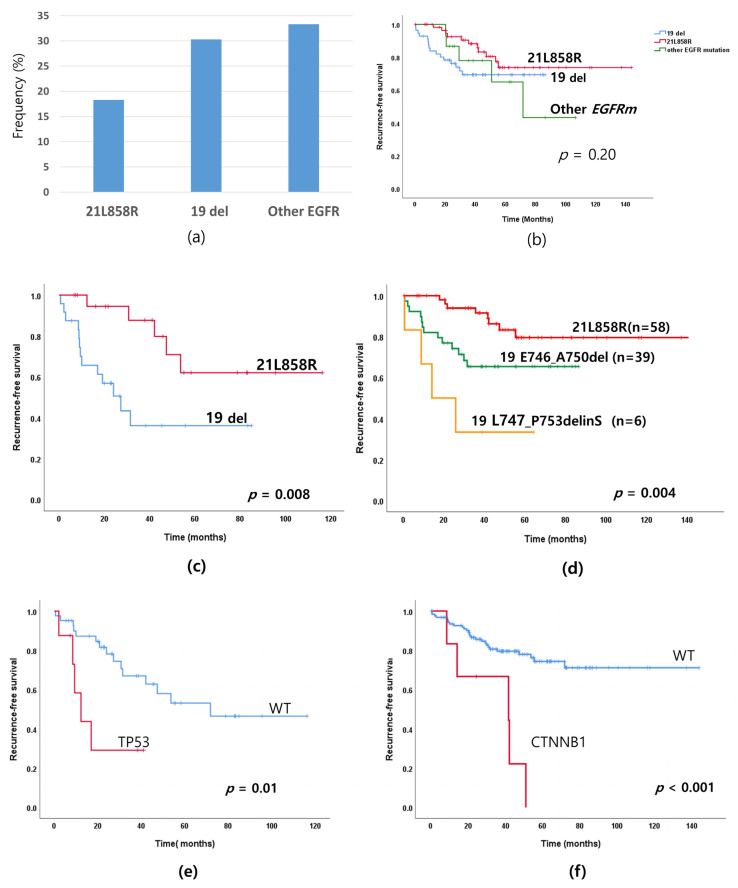
Recurrence-free survival (RFS) of *EGFR* mutation subtype in EGFR mutated lung adenocarcinoma. (**a**) Recurrence rate according to *EGFR* mutation types. (**b**) RFS according to *EGFR* mutation types in all cancer stages. (**c**) RFS according to *EGFR* mutation type in stage IB–IIIA cancer. (**d**) RFS according to the three most frequent *EGFR* mutations (E746_A750del, L747_P753delinS of 19-Del mutation, and 21L858R) in whole patients. (**e**) RFS according to the TP53 mutation in stage IB–IIIA cancer. (**f**) RFS according to the *CTNNB1* mutation.

**Table 1 cancers-13-03632-t001:** Comparison of the clinic-pathological characteristics of 131 patients with EGFR-mutated lung adenocarcinoma according to recurrence.

Characteristics	All Patients	No Recurrence (%)	Recurrence (%)	*p*-Value
**The number of patients**	131	98 (74.8)	33 (25.2)	*p* < 0.001
**Age**	64.0 ± 11.2	65.5 ± 10.9	58.8 ± 11.2	0.01
<65 years	71 (54.2)	47 (66.2)	24 (33.8)	0.007
≥65 years	60 (45.8)	51 (85.0)	9 (15.0)	
**Sex**				0.7
male	57 (43.8)	43 (75.5)	14 (24.5)	
female	74 (56.2)	55 (74.3)	19 (25.7)	
**Smoking history (*n* = 128 ^a^)**				0.73
Non-smoker	80 (62.5)	60 (75.0)	20 (25.0)	
Ever-smoker	48 (37.5)	36 (75.0)	12 (25.0)	
**Smoking dose**	9.6 ± 16.5	10.1 ± 16.9	8.1 ± 14.9	0.56
Non-smoker	80 (62.5)	60 (75.0)	20 (25.0)	0.47 *
<10 PYS	16 (12.6)	13 (81.2)	3 (18.8)	
11–30 PYS	12 (10.2)	10 (83.3)	2 (16.7)	
>30 PYS	20 (15.7)	14 (75)	6 (25)	
**Stage ^b^**				0.001 *
IA	82 (62.6)	70 (85.4)	12 (14.6)	
IB	29 (22.1)	23 (82.8)	5 (17.2)	
II	9 (6.9)	3 (33.3)	6 (66.7)	
IIIA	11 (8.4)	2 (18.2)	9 (81.8)	
**Surgical procedure**				0.53
sublobar resection	20 (15.4)	14 (70)	6 (30.0)	
lobectomy	111 (84.6)	85 (76.6)	26 (23.4)	
**Pathologic invasion**				
VPI	28 (20.0)	13 (46.4)	15 (53.6)	0.001
No VPI	103 (80.0)	87 (84.3)	16 (15.7)	
LVI	8 (5.3)	2 (25)	6 (75.0)	0.003 *
no LVI	123 (94.7)	98 (78.0)	25 (22.0)	
**EGFR mutation subtype**				
L858R/L861Q	58/2 (45.8)	49 (81.7)	11 (18.3)	0.3 *
Exon 19 del	56 (42.7)	39 (69.7)	17 (30.3)	
Exon 20 ins	5 (3.8)	3 (60)	2 (40)	
Compound mutation ^c^	10 (7.7)	7 (66.7)	3 (30)	
**Adjuvant chemotherapy**				
No treatment	94 (71.8)	79 (85.1)	15 (14.9)	
Treatment	37 (28.2)	19 (51.4)	18 (47.2)	0.001
**Death**	16 (12.3)	0 (0)	16 (100)	<0.001 *

Abbreviation: visceral-parietal invasion:VPI, lymphvascular invasion: LVI. ^a^ 128 of the 131 patients had smoking histories indicated in the medical chart records. ^b^ Pathologic stage was determined according to the American Joint Committee on Cancer (8th edition). ^c^ Compound mutation: Two or more EGFR mutations in the same tumor, for example, G719X + S768I. * We marked an asterisk (*) by the *p*-value when the Fisher’s exact test was used.

**Table 2 cancers-13-03632-t002:** Prognostic factors for recurrence based on multivariate analysis of resected EGFR-mutated lung adenocarcinoma.

		Univariate Analysis			Multivariate Analysis		
Category	Variables	HR	95% CI	*p*-Value	HR	95% CI	*p*-Value
Age	≥65 vs. <65	0.37	0.16–0.82	0.009	0.38	0.15–0.97	0.04
Sex	Male vs. female	0.93	0.46–1.89	0.85	0.75	0.24–2.28	0.61
Smoking history	Ever-smoker vs. non-smoker	1.04	0.49–2.17	0.91	1.15	0.36–3.59	0.8
Pathologic stage	II-III vs. I	9.23	4.45–18.7	<0.001	8.02	3.73–17.2	<0.001
Extension of surgery	sublobar resection or lobectomy	1.32	0.54–3.21	0.53	1.44	0.74–2.80	0.27
Pathologic invasion	VPI vs. none	4.52	2.23–9.18	<0.001	8.02	3.73–17.2	<0.001
Adjuvant chemotherapy	Adjuvant chemothrapy	4.78	2.36–9.67	<0.001	2.21	1.14–4.24	0.017
No. of co-mutation	1 vs. 0	1.51	0.66–3.4	0.32	1.76	0.72–4.3	0.213
No. of co-mutation	2 vs. 0	3.84	1.47–10.1	0.006	5.57	1.57–19.6	0.008
EGFR mutation	19-del vs. 21L858R (all stage)	1.88	0.87–4.07	0.1	1.82	0.76–4.32	0.17
	19-del vs. 21L858R (IB-IIIA)	4.01	1.41–11.3	0.009	8.31	1.98–34.8	0.004
	pE746_A 750del vs. 21L858R	2.19	1.04–4.60	0.03	1.7	0.75–3.83	0.2
	pL747_P753delinS vs. pE746_A 750del	2.52	0.71–8.88	0.15	1.69	0.75–3.83	0.2
	pL747_P753delinS vs. 21L858R	6.76	1.93–23.7	0.003	7.55	1.91–29.8	0.004
*CTNNB1*	*CTNNB1/EGFR* vs. *EGFR*	5.19	1.97–13.6	0.001	8.65	3.0–24.9	<0.001
*TP53* mutation	*TP53/EGFR* vs. *EGFR* (All stage)	1.89	0.71–5.05	0.2	2.14	0.84–5.43	0.1
	*TP53/EGFR* vs. *EGFR*(IB-IIIA)	3.48	1.21–10.0	0.01	3.06	1.04–11.7	0.05

Abbreviations: CI, confidence interval; HR, hazard ratio; EGFR, epidermal growth factor receptor; VPI, visceral-pleural invasion, PD: poor differentiation, MD; moderate differentiation, WD; well differentiated; No, number. *p* values were calculated using multivariate Cox proportional hazard models, adjusted for age, sex, smoking status, and stage.

## Data Availability

The data presented in this study are available from the corresponding author upon reasonable request.

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
