# Peer review of "Targeted Next-Generation Sequencing Analysis Predicts the Recurrence in Resected Lung Adenocarcinoma Harboring EGFR Mutations"

_cancers, 2021, doi:10.3390/cancers13143632_

Round 1

Reviewer 1 Report

I appreciate that the authors have responded to the reviewers' comments.

Author Response

Reviewer 1 mentioned that he/she appreciates that we have responded to the reviewers' comments and he/she did not comment any further on our manuscript signing his review report. We appreciate him/her for his review.

Reviewer 2 Report

Thank you for your invitation for reviewing the article: Targeted next-generation sequencing analysis predicts the recurrence in resected lung adenocarcinoma harboring EGFR mutations.

This research paper is interesting and highly relevant for exploring underlying genomic profiles in a NSCLC patients cohort harboring pathogenic EGFR mutations, that may benefit patients with recurrent disease. In this study, the authors found that number of co-mutations and mutations of CTNNB1 are negatively associated with recurrence-free survival. The discussion section is concise, hence also comments on pros and cons of the author’s research shows reflection on the research conducted. Nevertheless, there are some points that need attention and is necessary for a thorough understanding of its relevance – and important clinical research conducted in this article. The comments are outlined below (or attached above to the authors).     

Comments:

  1. Line 63-64: The author’s mention: “…in order to identify driver oncogenes.”. I suggest revision of this sentence (driver mutations and driver oncogenes are of different meaning). A onco-gene is a consequence of a proto-oncogene being turned on or activated uncontrolled. Thus, leading to cancer. However, a tumor-suppressor-gene can also cause cancer, by its function being inactivated or turned off. In figure 1 is listed TP53, APC, RB1, BRCA1, BRCA2, PTEN, - which are also clinically relevant genes, despite being classified as tumor-suppressor genes. However, driver mutations can be the leading cause to uncontrolled cell growth/survival introduced in both tumor-suppressor genes and proto-oncogenes.
  2. Line 115-117 and supplementary method: The authors mention that variants with an allele frequency below 2% were excluded and the above were regarded as significant actionable mutations and used for further analysis. It could be interesting to known briefly how was this cut-off determined? And additionally, what was the accepted cut-off for phred-score/accuracy and coverage for a variant identified with low allele frequency?
  3. In continuation of comment 3: The authors are rightfully taking artifactual variants into account for analysis. From the supplementary method: The authors are focusing on oxoG (G:C > T:A) artifactual variants, which is associated with genomic library prep. However, under section 2 NGS processing, it can be found that NGS was conducted on DNA derived from FFPE. Deamination of cytosine is a commonly observed signature introduced into DNA from FFPE, known to cause C>T and G>A transitions. It has been shown that these artifactual false positives are predominately observed in variants with low allele frequency, hence accepting allele frequencies above 2% might challenge variant validity. Have the authors considered potential artifactual false positive discoveries from cytosine deamination? For instance, we have experienced that identifying cut-off values for allele frequencies by inspecting base substitutions as a function of allele ratios below the 1st quartile of the mean allele frequency of the sample has provided a reasonable alternative.
  4. Line 192-198: I would suggest revision of this paragraph. I fully agree to the purpose of investigating patient of stage I cancer, however – the drawn conclusion of 17 stage I patients experiencing recurrence contrary 15 stage II-IIIA patients experiencing recurrence is slightly biased, as the authors are comparing subgroups of 111 and 20, respectively. Hence, referring to proportions, as the authors have on previous sentence, is for suitable.
  5. Line 226 – 228: It is mentioned that stage IA patients with low recurrence rate were excluded, due to the RFS difference between 19-del and 21-L858R were obvious. Thus, only focusing on stage IB. Nonetheless, it can be seen from Table 1, that the recurrence rate for stage IA and IB are similar, 14.6% and 17.2%, respectively. Furthermore, as shown on the respective figure, the RFS difference between stages IA and II-III seems to be as clear. To that end, the rationale for removing IA from the analysis (or not removing IB instead) should be described.
  6. To figure 3c-d: It would be easier for the reader to follow if 21-L858R were to be the same color in the two plots and then re-color 19E746_A750del and 19L747_P753 to specific colors not used 3c.
  7. Line 223-225: Frequencies in the text do not match frequencies represented by the bar-chart in figure 3a. The recurrence frequency in the text is 30.3% for 19-del, 18.6% for 21L858R and 33.3% for other EGFR However, on the bar-plot is represented approximately 34% for 19-del, above 20% for 21L858R and above 35% for other EGFR-mutations. This is making the text confusing.    
  8. Figures: The Kaplan-Meier graphs and oncoplot looks streamline and thoroughly presented for the eye and somehow easy to follow with color codes (comment 6).     
    1. I think if Figure 1a is needed, it could be more strongly presented by a percentual pie-chart otherwise I would recommend to introduced it as a horizontal bar chart on the oncoplot in Figure 1b. Or completely leave it out, as the oncoplot tells the story.
    2. The bar-charts need revision. It would be recommended to choose on way of representing them, for instance with or without 3D effect. Centering of the y-label in center is nicely done for all Kaplan-Meier plots, but a bit off track in Figure 3a. Moreover, Figure 1a is missing the y-label of frequency.
    3. In Figure 3d the number of patients is annotated to the plot. From the oncoplot it is possible to count 39 19del patients, hence 6 is left out from the oncoplot when compared to Figure 3d. I suggest that the authors make it clear which patients are included in the oncoplot.

Minor comments:

  1. There are some typos throughout the text. Ex: Line 197: “II-IIA” Line 200: “DFS
  2. Line 98: “…manufacture’s protocol” – Add which protocol this is.
  3. Line 362-368: The nomenclature of human gene names should be written in capital and in italics.

Author Response

This manuscript is a resubmission of an earlier submission. The following is a list of the peer review reports and author responses from that submission.

Round 1

Reviewer 1 Report

In this article, the authors assessed the genetic profile in patients with lung adenocarcinoma harboring EGFR mutations. They hypothesized that in those subjects harboring EGFR mutations the co-occurrence of other mutations would associate with poorer prognosis, in regard to recurrence-free survival (RFS). Their findings showed that there were some factors that contribute to shorter RFS, such as the number of concomitant mutations, mutations on CTNNB1, and specific mutations on exon 19 of EGFR. The clinical question is well-reasoned, and the experimental design and execution is overall suitable. Nonetheless, there are some points of concern that would be necessary to be addressed in order to elevate the results and discussion of the current manuscript, as follows:

  1. It is very interesting that the authors opted for the use of a targeted panel for their NGS method, instead of WGS or WES. In a clinical setting targeted panels are widely used given their versatility, cost, and relatively fast analysis. So, in the future, these findings could be adapted to the clinical routine in a more seamless manner. Nonetheless, the methodology is scarcely provided by the authors (section 2.3 NGS data analysis). That being said, and for a better overall understanding about the methodology, it would be interesting to have a workflow pipeline with all steps included added as a figure/table or supplementary information;
  2. In connection with the previous point, the authors mentioned that “Matched germline DNA of the patients, which is normal control for mutation analysis, was unavailable for this retrospective study” (lines 106-07). This is unfortunately a common reality in the clinic, and there are many ways to reduce the absence of such controls in the analysis. Thus, by providing a further picture of the methodology, as suggested above, that would benefit the manuscript.
    Moreover, the authors mentioned that the NGS pipeline is well-established and has been validated (line 319), referring to Gradishar et al, 2020. However, such work refers to the clinical practice guidelines in breast cancer and it does not provide information about such NGS pipeline. Please, revise the literature references and provide adequate information in the body of the manuscript;
  3. The clinical and pathological parameters were evaluated by chi-squared or Fisher exact test (lines 131-32). The authors should clarify the criteria for using one or the other, given that the sampling size is large enough for performing chi-squared test, and the dimensions of the contingency tables seem to be variable;
  4. Although the authors are clear on their experimental design in regarding only EGFR-mutated cases, it would be very interesting to have a few cases with non-mutated EGFR for a few reasons, such as (i) to show that the current cohort actually reflects the fact that, in general, EGFR mutation leads to a better prognosis in lung adenocarcinoma cases (lines 44-45); and (ii) whether EGFR is a potential driver gene (more details below);
  5. In regard to whether EGFR might be a potential driver gene in those patients, the inclusion of non-mutated EGFR (EGFRwt) cases would allow for the comparison of (i) RFS of EGFRwt and EGFRL858R. It would be interesting to see whether those frequencies are similar. If so, could EGFRL858R be a potential neutral mutation, instead of conferring better prognosis? And (ii) how would co-occurring mutations compare on both groups? Would the number of concomitant mutations in EGFRwt be similar to that of EFGRmut (Fig. 2b)?
  6. Why does the K-M curve (censored cases) for EGFRL858R look different in Figure 3c, in comparison to figure 3b and d? The authors mentioned briefly that “in subgroup analysis of the patients with stage IB-IIIA group” (lines 211-12) and in the figure 3c legend. At first, I believed that all stage IA cases were removed (n= 82). However, figure 3d legend also refers to stages IB-IIIA and it shows a different curve for EGFRL858R. It will be important for the authors to clarify that.

Minor points:

  1. Line 160 mentions recurrence frequency in patients that did not receive chemotherapy treatment was 14.9%. However, in Table 1 that frequency is of 15.9%. Please, correct.
  2. It is mentioned that the number of recurrent stage I patients was higher than that for stages II-IIIA, 17 and 15, respectively (lines 179-82). However, when looking at frequencies those numbers are more informative, 15.3% (17/111) and 75% (15/20), respectively. Thus, instead of using absolute numbers, frequencies are more suitable in this case.
  3. Lines 208-10 are a bit confusing due to the reverse order of recurrence rates. Given that the authors mentioned that “Relapse rates increased the most at 18.6% (11/60) for EGFR… and 33.3% (5/15) for other EFGR mutations”, it would be more appropriate to present the data in a decreasing manner.
  4. By convention, human gene names should be presented as capital letters and in italic.
  5. Overall revision of the text. For example, some typos or phrasing: line 305, “pE746??_A750del”; line 51, “an important unmet needs”; line 124, “Secondary primary lung cancer”; line 202, “RFS times became shorter”.

Reviewer 2 Report

The authors reported on specific risk factors for postoperative recurrence based on comprehensive genetic analysis using NGS.
Although their point of view is interesting, it is not novel and lacks statistical power.
I would like to comment on the following points.
1. The observation period for postoperative recurrence of EGFR mutation-positive lung cancer for each case is not consistent. Some cases may recur in the future, and this may be regarded as an arbitrary observation period. Is it possible to make improvements in order to clear this point?
2. You conclude that there are many cases of recurrence in patients with CTNNB1 mutations, but the number of cases is too small. Can these trends be reproduced in public databases, etc.?
3. Isn't the description in lines 285-288 insufficient to explain why postoperative recurrence is common in patients with CTNNB1 mutations?
4. Can you please include in the discussion section a discussion of the literature on what treatment strategies should be considered for patients with CTNNB1 mutations?

Round 2

Reviewer 2 Report

I think the authors have responded appropriately to the reviewers' suggestions.

I understand the treatment of post-operative cases that have not recurred, but I still have questions about whether this is an appropriate statistical method when discussing the risk of recurrence.